# Sentence-level Prompts Benefit Composed Image Retrieval

**Yang Bai**[1]  **Xinxing Xu**[1]  **Yong Liu**[1]  **Salman Khan**[2,3]  **Fahad Khan**[2]
**Wangmeng Zuo**[4]  **Rick Siow Mong Goh**[1]  **Chun-Mei Feng**[1]*

[1] Institute of High Performance Computing (IHPC),
Agency for Science, Technology and Research (A*STAR), Singapore
[2] Mohamed bin Zayed University of Artificial Intelligence (MBZUAI), UAE
[3] Australian National University, Canberra ACT, Australia
[4] Harbin Institute of Technology, Harbin, China
fengcm.ai@gmail.com
https://github.com/chunmeifeng/SPRC

## Abstract

Composed image retrieval (CIR) is the task of retrieving specific images by using a query that involves both a reference image and a relative caption. Most existing CIR models adopt the late-fusion strategy to combine visual and language features. Besides, several approaches have also been suggested to generate a pseudo-word token from the reference image, which is further integrated into the relative caption for CIR. However, these pseudo-word-based prompting methods have limitations when target image encompasses complex changes on reference image, *e.g.,* object removal and attribute modification. In this work, we demonstrate that learning an appropriate sentence-level prompt for the relative caption (SPRC) is sufficient for achieving effective composed image retrieval. Instead of relying on pseudo-word-based prompts, we propose to leverage pretrained V-L models, *e.g.,* BLIP-2, to generate sentence-level prompts. By concatenating the learned sentence-level prompt with the relative caption, one can readily use existing text-based image retrieval models to enhance CIR performance. Furthermore, we introduce both image-text contrastive loss and text prompt alignment loss to enforce the learning of suitable sentence-level prompts. Experiments show that our proposed method performs favorably against the state-of-the-art CIR methods on the Fashion-IQ and CIRR datasets.

## 1 Introduction

Composed image retrieval (CIR) (Liu et al., 2021; Vo et al., 2019; Baldrati et al., 2022b) is a challenging image retrieval task where the query involves both a reference image and a relative caption. Concretely, it aims at retrieving the target image which encompasses the specified changes provided by the relative caption while otherwise preserving the visual similarity to the reference image. Due to the bimodal nature of the query, CIR can provide more precise control to describe the target image, and allows interactive retrieval by changing the reference image and relative caption based on the current retrieval results. Consequently, these advantages make CIR appealing in several applications such as e-commerce and internet search.

Despite its wide applications, the bi-modality of the query makes CIR very challenging. In contrast to text-to-image and image-to-image retrieval, the core of CIR is to compose the information from the reference image and relative caption into a joint embedding for computing similarities with the candidate images. To this end, the prevalent CIR methods usually adopt the late-fusion strategy (Anwaar et al., 2021; Chen et al., 2020; Dodds et al., 2020; Liu et al., 2021; Vo et al., 2019), which first extracts features from the reference image and relative caption, and then combines them to form the fused feature. Nonetheless, the target image may encompass complex changes on the reference image, such as removing some objects, modifying object/scene attributes, or adding new

---

*Corresponding author.

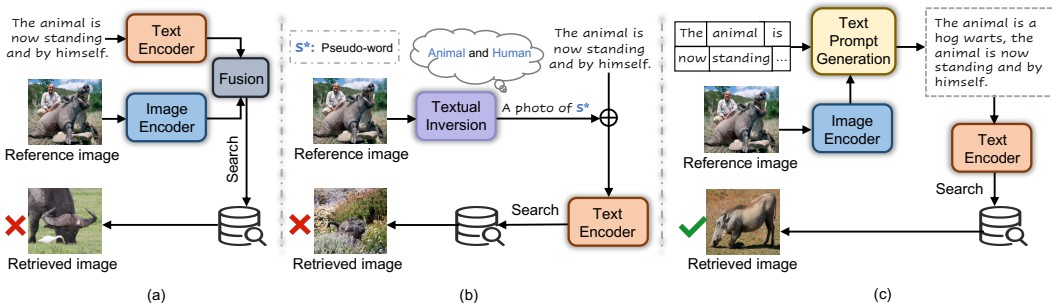

Figure 1: **Workflows** of existing CIR methods and Ours: (a) Late fusion, (b) pseudo-word embedding, and (c) our proposed method. It can be seen that late fusion and pseudo-word embedding are limited in handling the cases where multiple objects are involved in the reference image and complex changes, *e.g., object removal* or *attribute modification*, are included in the relative caption. In comparison, our proposed method is suggested to generate a *sentence-level prompt* from the reference image and relative caption. Using an accurate sentence-level prompt, *e.g.,* "The animal is a hog war", one can readily use existing text-based image retrieval models to correctly retrieve the target image.

objects. Existing late-fusion methods (Anwaar et al., 2021) can be effective in handling the cases of adding new objects, but are still limited in retrieving the target images relevant to object removal or attribute modification of reference images (see Fig. 1(a)).

Recently, several zero-shot CIR methods based on pseudo-word embedding have been suggested for zero-shot CIR. In these methods, textual inversion (Gal et al., 2022) or CLIP-based mapping (Baldrati et al., 2022b) are usually adopted to transform reference image into its pseudo-word embedding. Then, simple integration is adopted to combine pseudo-word token with relative caption, and existing text-to-image retrieval methods, *e.g.,* CLIP (Radford et al., 2021), can be leveraged to retrieve target images. However, the pseudo-word token is an overall textual representation of the objects in the reference image, and the length of the pseudo-word token is small. When there are multiple object categories in a reference image, the above methods generally fail to decouple these objects in the pseudo-word tokens. Once object removal or attribute modification of reference image is involved in the target image, it usually cannot be correctly reflected in the integration process of pseudo-word token and relative caption. Consequently, these methods perform poorly when multiple objects are involved in the reference image and complex changes, *e.g.,* object removal or attribute modification, are included in the relative caption (see Fig. 1(b)).

In this work, we suggest to circumvent CIR from a new perspective. Using Fig. 1(c) as an example, the reference image show a man sits around a hog warts lying on the ground and stone. And the relative caption is "*the animal is now standing and by himself*". Thus, to match with the target image, complex changes are required to be applied to the reference image, including removing the man and adjusting the pose of hog warts. Interestingly, we find that the complex CIR task can be well addressed by adding a proper sentence-level prompt to relative caption (SPRC). For the example in Fig. 1(c), the sentence-level prompt can be "*The animal is a hog warts*", which is then concatenated with the relative caption to form a new text query well matched with the target image. Thus, we argue that CIR can be effectively solved by learning proper sentence-level prompt.

We further present an effective method for learning sentence-level prompt. In pseudo-word embedding based methods, the pseudo-word token solely depends on reference image and is of short length. In contrast, **sentence-level prompt** has a **richer expressivity** and should depend on both the reference image and relative caption, thereby yielding **precise descriptions of specific elements** in a reference image that described in the relative caption. We adopt a lightweight Querying Transformer, *i.e.,* Q-Former (Li et al., 2023), as the text prompt generation network, which takes relative caption and reference image feature as input to generate sentence-level prompt. By concatenating the learned sentence-level prompt with relative caption, one can readily use existing text-based image retrieval models to facilitate CIR. Two complimentary loss terms are introduced to facilitate the learning of Q-Former for generating proper sentence-level prompts. First, we adopt an image-text contrastive loss to ensure that the concatenation of sentence-level prompt and relative caption is effective in retrieving the target image. Second, a **text prompt alignment loss** is further introduced to directly guide the learning of sentence-level prompt generation. Specifically, we leverage the optimization-based method to obtain an auxiliary text prompt, and we then enforce the output of Q-Former to be aligned with the auxiliary text prompt. Experiments conducted on the **Fashion-IQ** and **CIRR** datasets demonstrate that our method achieves state-of-the-art results on both datasets. In a nutshell, our contributions are summarized as follows:

- We present a new method to tackle CIR by learning proper sentence-level prompts derived from both the reference image and relative caption, which offer *enhanced expressiveness and precision*, improving CIR effectiveness.
- Two loss terms, *i.e.,* image-text contrastive loss and text prompt alignment loss, are proposed to enhance the training of the text prompt generation network, with the latter *guiding* the alignment of sentence-level prompt generation to an auxiliary text prompt.
- Experiments on the Fashion-IQ and CIRR datasets show that our proposed method performs favorably against the state-of-the-art CIR methods.

## 2  RELATED WORK

**Composed Image Retrieval.** The prevalent contemporary CIR methods primarily employ the late fusion paradigm to collectively incorporate the characteristics of the reference image and relative caption (see Fig. 1 (a)). Then the fused feature is compared with those of all candidate images to determine the closest match (Anwaar et al., 2021; Chen et al., 2020; Dodds et al., 2020; Liu et al., 2021; Vo et al., 2019). Various feature fusion mechanisms (Vo et al., 2019) and attention mechanisms (Chen et al., 2020; Dodds et al., 2020) have exhibited prominent performance in CIR. Subsequently, capitalizing on the robust capabilities of pre-trained models, a number of CIR methods that adeptly amalgamate image and text features extracted from autonomously trained visual and text encoders (Baldrati et al., 2022a; Goenka et al., 2022; Ray et al., 2023; Liu et al., 2023a). Another category of approaches suggest to transform reference image into its pseudo-word embedding (Saito et al., 2023; Baldrati et al., 2023), which is then combined with relative caption for text-to-image retrieval. However, the pseudo-word embeddings learned through such methods are often limited in disentangling multiple objects in the reference images, as shown in Fig. 1 (b). When the reference image involves many objects and many intricate attributes, and the relative caption only refers to part of them, the pseudo-word embeddings generally cannot remove irrelevant objects or attributes. Liu *et al.* created text with semantics opposite to that of the original text and add learnable tokens to the text to retrieve images in two distinct queries (Liu et al., 2023b). Nevertheless, this approach to fixed text prompt learning fails to modify the intrinsic information within the relative caption itself, thereby constraining retrieval performance. Concurrent to our work, CIReVL Karthik et al. (2023) used frozen captioning models to generate captions from reference images to complement relative captions for training free CIR. Alternatively, our aim is to utilize both reference image and relative caption to learn sentence-level prompts, thereby enhancing the relative caption itself. This enables *precise descriptions to specific elements* in the reference image that are pertinent to the relative caption while *avoiding any interference from extraneous elements*, *e.g.,* irrelevant background and other objects.

**Vision-Language Foundation Models.** Thanks to the massive training data consisting of billions of image-text pairs, large-scale visual language models, *e.g.,* CLIP, have exhibited the ability to flexibly accommodate a range of downstream tasks without the need for specific task-related training data. Subsequently, significant advances have been made in the development of V-L foundation models. And these methods have demonstrated impressive performance across various vision tasks (Radford et al., 2021; Zhou et al., 2022a; Mokady et al., 2021; Song et al., 2022). Naturally, V-L foundation models can also be used in the CIR methods (Baldrati et al., 2022a; Goenka et al., 2022; Devlin et al., 2018; Ray et al., 2023). Concretely, existing late fusion-based CIR methods fuse features extracted using visual and textual encoders, while the methods based pseudo-word embedding leverage the text encoder in CLIP to learn pseudo-word token through textual inversion (Saito et al., 2023; Baldrati et al., 2023). In contrast, we incorporate both the image and text encoders of V-L foundation models to learn sentence-level *dynamic prompt* that describes the required visual characteristics of reference image while being consistent and complementary with relative caption.

**Text Prompt Tuning for Vision-Language Models.** To facilitate the applications of large-scale pre-trained V-L models to downstream tasks, prompt tuning serves as an effective way of adapting pre-trained models to each task in a parameter-efficient manner. This methodology has yielded enhanced performance across a spectrum of tasks within the domains of natural language processing (Devlin et al., 2018; Radford et al., 2018) and computer vision (Jia et al., 2021; 2022; Feng et al., 2023c;a). Prompts can be added to either text encoder (*i.e.,* text prompt) or image encoder (*i.e.,* visual prompt), and only text prompt is considered in this work. After prompt tuning, prompts can be deployed in either static (Zhou et al., 2022b; Sun et al., 2022; Derakhshani et al., 2022; Khattak et al., 2023), or dynamic (Zhou et al., 2022a; Wang et al., 2022; Dong et al., 2022; Lu et al., 2022; Feng et al.,

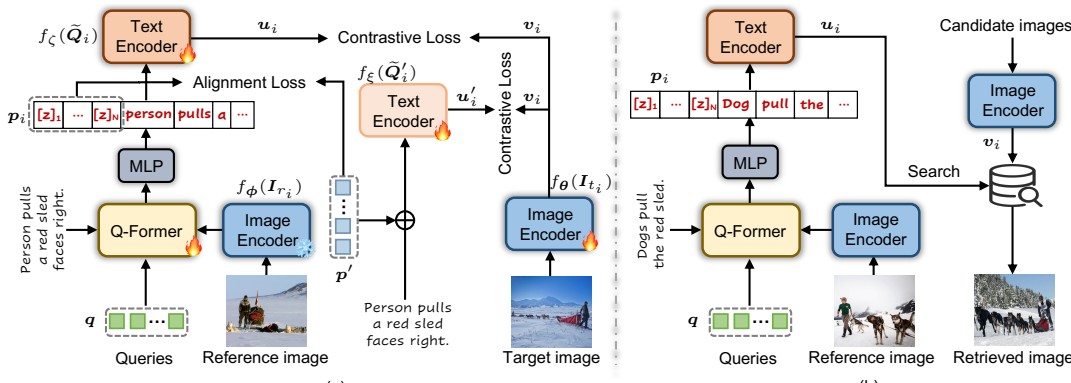

Figure 2: **Overall pipeline** of our proposed SPRC. **(a)** Illustration of the training phase, where proper sentence-level prompts are integrated into the relative captions thereby providing essential textual prompts containing information about the objects depicted in the reference images and their relative captions. Even in cases where the reference images involve multiple objects and the relative captions encompass complex modifications, these sentence-level prompts guarantee that the retrieval target image can be correctly reflected. **(b)** Illustration of the test phase, the learned sentence-level prompts are used to enhance the relative caption for text-image retrieval.

2023b) manners. In particular, static prompt is fixed all test samples, while dynamic prompt is sample-adaptive. In CIR, existing prompt tuning-based method, *i.e.,* (Liu et al., 2023a), adopt static prompt. As for our proposed method, the sentence-level prompt is introduced as a modified description of reference image to be consistent with relative caption, and thus should be dynamic.

## 3 METHODOLOGY

**Overview.** Denote $\{I_r, t\}$ by a composed query, where $I_r$ denotes the reference image, and $t$ denotes the relative caption. Composed image retrieval (CIR) aims at retrieving the target image $I_t$ among all candidates in an extensive image corpus $\mathcal{D}$. Specifically, the target image should encompass the specified changes provided by relative caption while preserving the other visual similarity to reference image, making CIR highly relevant but more challenging than text-to-image retrieval. In this work, we present a new perspective for CIR by transforming reference image to a proper sentence-level prompt. Consequently, sentence-level prompt is concatenated with relative caption, and existing text-to-image retrieval can then be used to tackle CIR. From Fig. 1(c), one can see that such a perspective is intuitively plausible. Once the sentence-level prompt "*The animal is a hog warts*" is generated, the concatenation of sentence-level prompt with relative caption "*the animal is now standing and by himself*" can be used to correctly retrieve the target image from image corpus $\mathcal{D}$. As shown in Fig. 2, our method is suggested to tackle CIR by learning sentence-level prompt to relative caption (SPRC), and involves two major modules, *i.e.,* sentence-level prompt generation and text-to-image retrieval. Moreover, to facilitate the learning of sentence-level prompt, we adopt both image-text contrastive loss and text prompt alignment loss. In the following, we will introduce the modules and learning objectives in more detail.

**Sentence-level Prompt Generation.** The reference image usually involves multiple objects as well as multiple attributes of objects or background. We note that the target image generally encompasses the removal of objects and the modification of attributes in reference image. Thus, in contrast to pseudo-word embedding (Saito et al., 2023) which directly learns a textual inversion of reference image, our generated sentence-level prompt generally has a longer length (*e.g.,* 32 tokens) with richer expressivity to describe the reference image while removing the necessary objects and attributes. Moreover, different from pseudo-word embedding, the sentence-level prompt generation module takes both reference image and relative caption as the input. In particular, we adopt a lightweight Querying Transformer, *i.e.,* Q-Former in BLIP-2, to build the text prompt generation module since it is effective in bridging the modality gap (Li et al., 2023). As shown in Fig. 2 (a), BLIP-2 image encoder $f_\phi(I_{r_i})$ is used to extract features from the reference image. Then, both the reference image feature and relative caption are fed into Q-Former. A set of learnable queries $q$ are also introduced for interacting with text and image features. In our method, the number of queries is set to keep the same as that of the sentence-level prompt. Finally, the output of Q-Former is fed into MLP layer to generate sentence-level prompt $p_i$.

**Text-to-Image Retrieval.** To begin with, the learned sentence-level prompt $\boldsymbol{p}$ is first concatenated with relative caption $\boldsymbol{t}$ to form an augmented text query $\widetilde{\boldsymbol{Q}}$. Once $\widetilde{\boldsymbol{Q}}$ is obtained, one can readily use existing text-to-image retrieval models to facilitate CIR. In general, we consider several representative methods, such as CLIP (Radford et al., 2021), BLIP (Li et al., 2022), and BLIP-2 (Li et al., 2023). In our implementation, we adopt BLIP-2 as the text and image encoder for text-to-image retrieval. This is attributed to the superior capabilities of BLIP over CLIP, as it represents a more powerful V-L pretraining network. Instead of training all model parameters from scratch, BLIP-2 (Li et al., 2023) leverages frozen pre-trained unimodal models for bootstrapping V-L pre-training. In contrast to BLIP, BLIP-2 (Li et al., 2023) introduces a lightweight query transformer, *i.e.*, Q-Former, to bridge the modality gap effectively. Further, we conduct ablation experiments to analyze the effect of text-to-image retrieval models on the performance of our approach, with details available in Table 3.

**Learning Objective.** Two loss terms are introduced to generate proper sentence-level prompt as well as boost CIR performance. The first loss term is image-text contrastive loss which has been widely adopted in existing text-to-image retrieval models (Radford et al., 2021; Li et al., 2022; 2023). Specifically, the image-text contrastive loss is defined as,

$$\mathcal{L}_c = -\frac{1}{\mathcal{B}} \sum_{i \in \mathcal{B}} \log \frac{\exp\left(\tau \boldsymbol{u}_i^T \boldsymbol{v}_i\right)}{\sum_{j \in \mathcal{B}} \exp\left(\tau \boldsymbol{u}_i^T \boldsymbol{v}_j\right)}, \tag{1}$$

where $\mathcal{B}$ denotes the batch size. For the $i$-th sample, $\boldsymbol{v}_i = f_{\boldsymbol{\theta}}(\boldsymbol{I}_{t_i})$ denotes the image feature from pretrained model, $\boldsymbol{u}_i = f_{\zeta}(\widetilde{\boldsymbol{Q}}_i)$ denotes the text feature of the augmented query $\widetilde{\boldsymbol{Q}}_i$. The minimization of $\mathcal{L}_c$ not only benefits the learning of text-to-image retrieval module, but also encourages the sentence-level prompt generation module to produce semantically meaningful and contextually coherent sentence-level prompt for enhancing retrieval. However, the image-text contrastive loss does not enforce direct constraint on the generated sentence-level prompt. So we further introduce the text prompt alignment loss to guide the learning of sentence-level prompt generation module,

$$\mathcal{L}_a = \|\boldsymbol{p}_i - \boldsymbol{p}'\|_2, \tag{2}$$

where $\boldsymbol{p}'$ denotes the auxiliary text prompt generated using an optimization-based process. In summary, our overall main branch (left branch as shown in Fig. 2 (a)) loss can be expressed as $\mathcal{L}_{main} = \mathcal{L}_c + \gamma \mathcal{L}_a$. Let $\boldsymbol{Q}'_i$ be the concatenation of $\boldsymbol{p}'$ and relative caption $\boldsymbol{t}_i$. We have $\boldsymbol{v}_i = f_{\boldsymbol{\theta}}(\boldsymbol{I}_i)$ and $\boldsymbol{u}'_i = f_{\xi}(\widetilde{\boldsymbol{Q}}'_i)$. Then, the auxiliary text prompt $\boldsymbol{p}'$ can be optimized by $\mathcal{L}_{p'}$,

$$\mathcal{L}_{p'} = -\frac{1}{\mathcal{B}} \sum_{i \in \mathcal{B}} \log \frac{\exp\left(\tau \boldsymbol{v}_i^T \boldsymbol{u}'_i\right)}{\sum_{j \in \mathcal{B}} \exp\left(\tau \boldsymbol{v}_j^T \boldsymbol{u}'_i\right)} . \tag{3}$$

Given $\boldsymbol{p}'$, we can use the text prompt alignment loss to directly guide the learning of sentence-level prompt generation module. The framework jointly optimizes the main branch and $\boldsymbol{p}'$, where the total loss function is defined as $\mathcal{L} = \mathcal{L}_{main} + \mathcal{L}_{p'}$. To sum up, text prompt alignment loss and image-text contrastive loss can be used to learn sentence-level prompt generation module from different perspectives. Thus, they are complementary and both benefit the learning of our proposed model, and the experimental result also verify their effectiveness.

# 4 EXPERIMENTS

## 4.1 EXPERIMENTAL SETUP

**Implementation Details.** Our method is implemented with Pytorch on one NVIDIA RTX A100 GPU with 40GB memory. We follow the design of BLIP-2 (Li et al., 2023), the visual and text encoders are initialized from the BLIP-2 pretrained model with ViT-G and AdamW (Loshchilov & Hutter, 2017) optimizer with a weight decay of 0.05. We resize the input image size to $224 \times 224$ and with a padding ratio of 1.25 for uniformity (Baldrati et al., 2022b). The learning rate is initialized to 1e-5 and 2e-5 following a cosine schedule for the **CIRR** and **Fashion-IQ** datasets, respectively. The hyperparameters of prompt length and $\gamma$ are set to 32 and 0.8, respectively.

**Datasets and Metrics.** We evaluate our method on two CIR benchmarks: (1) **Fashion-IQ** a fashion dataset with $77,684$ images forming $30,134$ triplets (Wu et al., 2021). We utilize Recall@K as the evaluation metric, which reflect the percentage of queries whose true target ranked within the top

Table 1: **Quantitative** comparison across competing methods on the **Fashion-IQ** validation set, where **Avg.** indicates the average results across all the metrics in the three different classes. The best and second-best results are marked in **bold** and underlined, respectively.

| | Dress | | Shirt | | Toptee | | Average | | |
|---|---|---|---|---|---|---|---|---|---|
| **Method** | **R@10** | **R@50** | **R@10** | **R@50** | **R@10** | **R@50** | **R@10** | **R@50** | **Avg.** |
| JVSM (Chen & Bazzani, 2020) | 10.70 | 25.90 | 12.00 | 27.10 | 13.00 | 26.90 | 11.90 | 26.60 | 19.26 |
| CIRPLANT (Liu et al., 2021) | 17.45 | 40.41 | 17.53 | 38.81 | 61.64 | 45.38 | 18.87 | 41.53 | 30.20 |
| TRACE w/BER (Jandial et al., 2020) | 22.70 | 44.91 | 20.80 | 40.80 | 24.22 | 49.80 | 22.57 | 46.19 | 34.00 |
| VAL w/GloVe (Chen et al., 2020) | 22.53 | 44.00 | 22.38 | 44.15 | 27.53 | 51.68 | 24.15 | 46.61 | 35.38 |
| MAAF (Dodds et al., 2020) | 23.80 | 48.60 | 21.30 | 44.20 | 27.90 | 53.60 | 24.30 | 48.80 | 36.60 |
| CurlingNet (Yu et al., 2020) | 26.15 | 53.24 | 21.45 | 44.56 | 30.12 | 55.23 | 25.90 | 51.01 | 34.36 |
| RTIC-GCN (Shin et al., 2021) | 29.15 | 54.04 | 23.79 | 47.25 | 31.61 | 57.98 | 28.18 | 53.09 | 40.64 |
| CoSMo (Lee et al., 2021) | 25.64 | 50.30 | 24.90 | 49.18 | 29.21 | 57.46 | 26.58 | 52.31 | 39.45 |
| ARTEMIS (Delmas et al., 2022) | 27.16 | 52.40 | 21.78 | 43.64 | 29.20 | 53.83 | 26.05 | 50.29 | 38.04 |
| DCNet (Kim et al., 2021) | 28.95 | 56.07 | 23.95 | 47.30 | 30.44 | 58.29 | 27.78 | 53.89 | 40.84 |
| SAC w/BERT (Jandial et al., 2022) | 26.52 | 51.01 | 28.02 | 51.86 | 32.70 | 61.23 | 29.08 | 54.70 | 41.89 |
| FashionVLP (Goenka et al., 2022) | 32.42 | 60.29 | 31.89 | 58.44 | 38.51 | 68.79 | 34.27 | 62.51 | 48.39 |
| LF-CLIP (Combiner) (Baldrati et al., 2022b) | 31.63 | 56.67 | 36.36 | 58.00 | 38.19 | 62.42 | 35.39 | 59.03 | 47.21 |
| LF-BLIP (Baldrati et al., 2022b) | 25.31 | 44.05 | 25.39 | 43.57 | 26.54 | 44.48 | 25.75 | 43.98 | 34.88 |
| CASE (Levy et al., 2023) | 47.44 | 69.36 | 48.48 | 70.23 | 50.18 | 72.24 | 48.79 | 70.68 | 59.74 |
| AMC (Zhu et al., 2023) | 31.73 | 59.25 | 30.67 | 59.08 | 36.21 | 66.06 | 32.87 | 61.64 | 47.25 |
| CoVR-BLIP (Ventura et al., 2023) | 44.55 | 69.03 | 48.43 | 67.42 | 52.60 | 74.31 | 48.53 | 70.25 | 59.39 |
| CLIP4CIR (Baldrati et al., 2022a) | 33.81 | 59.40 | 39.99 | 60.45 | 41.41 | 65.37 | 38.32 | 61.74 | 50.03 |
| BLIP4CIR+Bi (Liu et al., 2023a) | 42.09 | 67.33 | 41.76 | 64.28 | 46.61 | 70.32 | 43.49 | 67.31 | 55.04 |
| FAME-ViL[†] (Han et al., 2023) | 42.19 | 67.38 | 47.64 | 68.79 | 50.69 | 73.07 | 46.84 | 69.75 | 58.29 |
| TG-CIR (Wen et al., 2023) | 45.22 | 69.66 | 52.60 | 72.52 | 56.14 | 77.10 | 51.32 | 73.09 | 58.05 |
| DRA (Jiang et al., 2023) | 33.98 | 60.67 | 40.74 | 61.93 | 42.09 | 66.97 | 38.93 | 63.19 | 51.06 |
| Re-ranking (Liu et al., 2023b) | 48.14 | 71.43 | 50.15 | 71.25 | 55.23 | 76.80 | 51.17 | 73.13 | 62.15 |
| CompoDiff (Gu et al., 2023) | 40.65 | 57.14 | 36.87 | 57.39 | 43.93 | 61.17 | 40.48 | 58.57 | 49.53 |
| Ours (SPRC) | **49.18** | **72.43** | **55.64** | **73.89** | **59.35** | **78.58** | **54.92** | **74.97** | **64.85** |

$K$ candidates. Since the ground-truth labels for the test set of this dataset have not been publicly disclosed, we adopt the results on the validation set for performance evaluation. (2) **CIRR** is a general image dataset that comprises $36,554$ triplets derived from $21,552$ images from the popular natural language inference dataset NLVR2 (Suhr et al., 2018). We randomly split this dataset into `training`, `validation`, and `test` sets in an $8:1:1$ ratio. This dataset encompasses rich object interactions, addressing the issues of overly narrow domains and high number of false-negatives in the **Fashion-IQ** dataset, thereby allowing for a comprehensive evaluation of the effectiveness of our proposed method. We report the results of the competing methods on this dataset at different levels, *i.e.,* Recall@1, 5, 10, 50, and Recallsubset@K (Liu et al., 2021).

## 4.2 COMPARISON WITH STATE-OF-THE-ARTS

Table 1 summarizes the evaluation of various competing methods on the **Fashion-IQ** dataset. As can be seen from this table, our proposed method, *i.e.,* SPRC, achieves the highest recall across eight evaluation metrics on the three different classes. Specifically, compared with the second-best method, *i.e.,* Re-ranking (Liu et al., 2023b), SPRC improves the performance from $50.15$ to **55.64** in terms of the R@10 metric. More importantly, compared to BLIP4CIR+Bi (Liu et al., 2023a) that adds learnable tokens in the relative captions for CIR, our method still achieves significant improvements on class `Shirt`, *i.e.,* R@10: $41.76 \rightarrow$ **55.64**, and R@50: $64.28 \rightarrow$ **73.89**. This is mainly because BLIP4CIR+Bi (Liu et al., 2023a) learns a fixed text prompt fails to modify the inherent information of the relative captions themselves, thereby limiting its retrieval performance. In contrast, our SPRC leverages reference images and relative captions to learn sentence-level prompts for enhancing the relative captions themselves, thereby resulting in improved retrieval performance. It is worth noting that, SPRC achieves $4.3\%$ average performance gain against second-best method, *i.e.,* Re-ranking (Liu et al., 2023b). Nevertheless, the two-stage mechanism of Re-ranking leads to the increase of training and inference time. Notably, we find that modifying SPRC into a two-stage mechanism (termed SPRC$^2$) achieved higher **Avg.** performance, *i.e.,* Re-ranking (Liu et al., 2023b): $62.15$ *vs.* SPRC$^2$: **64.85**. To sum up, the results show that learning proper sentence-level prompts

Table 2: **Quantitative** comparison across competing methods on the **CIRR** test set, where **Avg.** indicates the average results across all the metrics in the three different settings, Recall$_s$@K indicates the Recallsubset@K. The best and second-best results are marked in **bold** and underlined, respectively.

| Method | Recall@K | | | | Recall$_s$@K | | | Avg. |
|---|---|---|---|---|---|---|---|---|
| | **K=1** | **K=5** | **K=10** | **K=50** | **K=1** | **K=2** | **K=3** | |
| TIRG (Vo et al., 2019) | 14.61 | 48.37 | 64.08 | 90.03 | 22.67 | 44.97 | 65.14 | 35.52 |
| TIRG+LastConv (Vo et al., 2019) | 11.04 | 35.68 | 51.27 | 83.29 | 23.82 | 45.65 | 64.55 | 29.75 |
| MAAF (Dodds et al., 2020) | 10.31 | 33.03 | 48.30 | 80.06 | 21.05 | 41.81 | 61.60 | 27.04 |
| MAAF-BERT (Dodds et al., 2020) | 10.12 | 33.10 | 48.01 | 80.57 | 22.04 | 42.41 | 62.14 | 27.57 |
| MAAF-IT (Dodds et al., 2020) | 9.90 | 32.86 | 48.83 | 80.27 | 21.17 | 42.04 | 60.91 | 27.02 |
| MAAF-RP (Dodds et al., 2020) | 10.22 | 33.32 | 48.68 | 81.84 | 21.41 | 42.17 | 61.60 | 27.37 |
| CIRPLANT (Liu et al., 2021) | 19.55 | 52.55 | 68.39 | 92.38 | 39.20 | 63.03 | 79.49 | 45.88 |
| ARTEMIS (Delmas et al., 2022) | 16.96 | 46.10 | 61.31 | 87.73 | 39.99 | 62.20 | 75.67 | 43.05 |
| LF-BLIP (Baldrati et al., 2022b) | 20.89 | 48.07 | 61.16 | 83.71 | 50.22 | 73.16 | 86.82 | 60.58 |
| LF-CLIP (Combiner) (Baldrati et al., 2022b) | 33.59 | 65.35 | 77.35 | 95.21 | 62.39 | 81.81 | 92.02 | 72.53 |
| CLIP4CIR (Baldrati et al., 2022a) | 38.53 | 69.98 | 81.86 | 95.93 | 68.19 | 85.64 | 94.17 | 69.09 |
| BLIP4CIR+Bi (Liu et al., 2023a) | 40.15 | 73.08 | 83.88 | 96.27 | 72.10 | 88.27 | 95.93 | 72.59 |
| CompoDiff (Gu et al., 2023) | 22.35 | 54.36 | 73.41 | 91.77 | 35.84 | 56.11 | 76.60 | 29.10 |
| CASE (Levy et al., 2023) | 48.00 | 79.11 | 87.25 | 97.57 | 75.88 | 90.58 | 96.00 | 77.50 |
| CASE Pre-LaSCo.Ca.[†] (Levy et al., 2023) | 49.35 | 80.02 | 88.75 | 97.47 | 76.48 | 90.37 | 95.71 | 78.25 |
| TG-CIR (Wen et al., 2023) | 45.25 | 78.29 | 87.16 | 97.30 | 72.84 | 89.25 | 95.13 | 75.57 |
| DRA (Jiang et al., 2023) | 39.93 | 72.07 | 83.83 | 96.43 | 71.04 | 87.74 | 94.72 | 71.55 |
| CoVR-BLIP (Ventura et al., 2023) | 49.69 | 78.60 | 86.77 | 94.31 | 75.01 | 88.12 | 93.16 | 80.81 |
| Re-ranking (Liu et al., 2023b) | 50.55 | 81.75 | 89.78 | 97.18 | 80.04 | 91.90 | 96.58 | 80.90 |
| Ours(SPRC) | 51.96 | 82.12 | 89.74 | 97.69 | 80.65 | 92.31 | 96.60 | 81.39 |
| Ours(SPRC$^2$) | **54.15** | **83.01** | **90.39** | **98.17** | **82.31** | **92.68** | **96.87** | **82.66** |

can provide a precise descriptions of specific elements in the reference image that described in the relative caption, while avoiding irrelevant interference, thereby benefiting retrieval.

We then evaluate the effectiveness of our method on a broader dataset, *i.e.,* **CIRR**, the test set results are summarized in Table 2. Compared with existing text prompt-based method BLIP4CIR+Bi (Liu et al., 2023a), consistent improvements of our method are attained on **CIRR**, *e.g.,* Recall@1: 40.15 → **51.96**, Recall@5: 73.08 → **82.12**, Recall@10: 83.88 → **89.74**, and Recall@50: 96.27 → **97.69**. Clearly, while BLIP4CIR+Bi (Liu et al., 2023a) introduces a learnable token into the relative captions, it fundamentally fails to enhance the intrinsic richness of the reference image. Consequently, its performance lags significantly behind our method. For Recall$_s$@K, our SPRC yields higher recall than BLIP4CIR+Bi (Liu et al., 2023a), *i.e.,* Recall$_s$@1: 72.10 *vs*. **80.65**, Recall$_s$@2: 88.27 *vs*. **92.31**, and 95.93 *vs*. **96.60**. Extensive evaluation across various metrics, *i.e.,* Recall@K and Recall$_s$@K, indicates that it is more desirable of learning dynamic sentence-level prompts than simply inserting a learnable token into the relative caption. Moreover, compared with the best competing method, Re-ranking (Liu et al., 2023b), our SPRC still achieves the highest recall. Regarding the metrics of Recall$_s$@K, SPRC obtains 0.61, 0.41, and 0.02 gains on three metrics. In particular, we find that SPRC$^2$, a variant of our method with Re-ranking (Liu et al., 2023b), further enhances the recall value, *e.g.,* **Avg.**: from 81.39 to **82.66**, and achieves **1.76** gains against the best competing method, Re-ranking (Liu et al., 2023b). The results are consistent with those on the **Fashion-IQ** dataset, demonstrating that our approach can effectively decouple complex objects in reference image using sentence-level prompts and better respond to relevant captions during the retrieval process.

## 4.3 ABLATION STUDIES

**Sentence-level Prompts *vs*. Late Fusion & Textual Inversion.** To assess the effectiveness of sentence-level prompt, we established three models, late fusion, textual inversion, and text prompt-based method, see details in Table 3. The performance of three CIR mechanisms are summarized in Table 3 under different retrieval modals, *i.e.,* CLIP, BLIP, and BLIP-2. In general, Late fusion mechanism performs worse than Text inversion, and Text inversion performs worse than Ours+BLIP-2 (Full.), which can be seen from the consistency results across three variants on CLIP, BLIP, and BLIP-2. Concretely, we observe three variants, BLIP-2-4CIR (Sum), Pic2Word (Saito et al., 2023)+BLIP-2, and Ours+BLIP-2 (Full.), perform better on BLIP-2 compared to CLIP and BLIP. This is mainly due to the fact that BLIP-2 performs better for text-to-image retrieval. Nonetheless, even when all

Table 3: **Ablation** studies with regard to three different CIR mechanisms on **CIRR** `validation` set, *i.e.,* (1) Late fusion: CLIP4CIR (Sum) (Baldrati et al., 2022b), BLIP4CIR (Sum), and BLIP-2-4CIR (Sum); (2) Textual inversion: variant Pic2Word (Saito et al., 2023) to supervised method with different retrieval modals, *i.e.,* CLIP, BLIP, and BLIP-2; (3) Text prompt-based method: BLIP4CIR+Bi (Liu et al., 2023a). ∗ indicates the results we reproduced.

| Method | CLIP | BLIP | BLIP-2 | Recall@K | | | | Recall$_s$@K | | | Avg. |
|---|---|---|---|---|---|---|---|---|---|---|---|
| | | | | K=1 | K=5 | K=10 | K=50 | K=1 | K=2 | K=3 | |
| CLIP4CIR (Sum) | ✓ | — | — | 32.62 | 67.02 | 79.74 | 95.31 | 65.41 | 84.67 | 92.54 | 66.21 |
| BLIP4CIR (Sum) | — | ✓ | — | 35.92 | 70.34 | 80.03 | 94.72 | 71.91 | 85.95 | 93.38 | 71.13 |
| BLIP-2-4CIR (Sum) | — | — | ✓ | 49.96 | 81.20 | 89.88 | 97.67 | 75.24 | 89.95 | 95.52 | 78.22 |
| Pic2Word$_{su}$+CLIP | ✓ | — | — | 33.94 | 64.17 | 74.52 | 90.05 | 73.57 | 89.47 | 95.09 | 68.87 |
| Pic2Word$_{su}$+BLIP | — | ✓ | — | 42.04 | 72.81 | 82.94 | 95.02 | 71.05 | 86.62 | 93.75 | 71.93 |
| Pic2Word$_{su}$+BLIP-2 | — | — | ✓ | 50.59 | 80.53 | 88.51 | 97.32 | 77.15 | 90.40 | 95.28 | 78.84 |
| BLIP4CIR+Bi* | — | ✓ | — | 42.46 | 75.61 | 84.73 | 96.89 | 72.81 | 88.41 | 96.04 | 74.21 |
| Ours+CLIP | ✓ | — | — | 41.61 | 75.05 | 84.83 | 96.58 | 71.61 | 87.63 | 94.59 | 73.33 |
| Ours+BLIP | — | ✓ | — | 49.05 | 79.72 | 88.02 | 97.61 | 77.68 | 91.53 | 96.24 | 78.70 |
| Ours+BLIP-2 (Full.) | — | — | ✓ | 54.39 | 84.76 | 91.25 | 97.99 | 81.27 | 93.30 | 97.20 | 83.02 |

(a) CIRR  (b) Fashion-IQ

Figure 3: **Qualitative** analysis of three different CIR mechanisms with respect to the Recallsubset@1 metric on two datasets, where the reference and target retrieval images are highlighted with green and red outline. Owing to limited space, we adjusted the dimensions of specific candidate images to achieve a visually appealing layout.

three mechanisms use BLIP-2 as the retrieval model, the performance of our `Ours+BLIP-2 (Full.)` is still significantly higher than the other three CIR mechanisms. This indicates that sentence-level prompts achieve superior performance, as they utilize reference images in conjunction with relative captions for handling complex cases in CIR, *e.g.,* object removal or attribute modification in multiple objects. Furthermore, even when employing the same retrieval model, BLIP-2, BLIP-2-4CIR+Bi (Sum) (Liu et al., 2023a) consistently achieve lower recall rates on all metrics in comparison to our method, with our approach attaining an average 6.09% improvement. This suggests that simply learning fixed text prompts are far from sufficient, and dynamic learning of sentence-level prompts tailored to relevant headlines often leads to improved performance, as shown in Tables 1 and 2.

Accordingly, we visualize the performance of the three CIR mechanisms and our method in Fig. 3, *i.e.,* BLIP-2-4CIR (Sum), variant Pic2Word (Saito et al., 2023)+BLIP-2, BLIP4CIR+Bi (Liu et al., 2023a), and `Ours+BLIP-2` (full), with respect to the R@1 metric on two datasets. A higher rank for the target images indicates better retrieval performance, and vice versa. When multiple objects are involved in the reference image and intricate modification like object removal or attribute modification are undertaken, these methods frequently encounter interference from other objects or the original attributes within the reference image.

**Discussion on Different Retrieval Modals.** Considering that various retrieval models can significantly affect the retrieval performance, in the final three rows of Table 3, we summarize the recalls of our method across different retrieval model configurations. It can be seen that a more powerful retrieval models lead to higher performance, *i.e.,* **Avg.**: `Ours+CLIP` = 73.33, `Ours+BLIP` = 78.70, and `Ours+BLIP-2` = 83.02. However, it is worth noting that `Ours+CLIP` outperforms CLIP4CIR

Table 4: **Ablation** studies with regard to different ways of text prompt generation, *i.e.,* (1) $\text{Ours}_{RC}$, (2) $\text{Pic2Word}_{su}$+BLIP-2 (Saito et al., 2023) and our proposed SPRC, where RC and RI indicate the relative caption and reference image, respectively.

| Method | RC | RI | Recall@K | | | | Recall$_s$@K | | | Avg. |
|---|---|---|---|---|---|---|---|---|---|---|
| | | | **K=1** | **K=5** | **K=10** | **K=50** | **K=1** | **K=2** | **K=3** | |
| $\text{Ours}_{RC}$ | ✓ | – | 39.76 | 70.92 | 80.84 | 94.96 | 71.65 | 86.48 | 93.84 | 71.29 |
| $\text{Ours}_{RI}$ | – | ✓ | 50.94 | 81.03 | 88.93 | 97.25 | 78.40 | 91.10 | 95.89 | 79.72 |
| Ours | ✓ | ✓ | 54.39 | 84.76 | 91.25 | 97.99 | 81.27 | 93.30 | 97.20 | 83.02 |

(Sum) and $\text{Pic2Word}_{su}$+CLIP on average by **7.12** and **4.46**, respectively, which are also CLIP-based CIR methods. Considering that the Q-Former in BLIP-2 can effectively reduce the modality gap, we adopt BLIP-2 as the retrieval model for our final full model, while the Q-Former also serves as the text prompt generation module in our method.

**Discussion on Prompt Length.** Here, we discuss the effect of the length of our core design, sentence-level prompt, on performance. Since the prompt length of the BLIP-2 pre-trained model is 32, we also set the prompts of different lengths, *i.e.,* 4, 8, 16, 32, and summarize the effect of these variations on our method in Fig. 4. As can be seen, as the prompt length increases, the model performance improves. However, longer lengths introduce slight higher complexity, *e.g.,* the FLOPs for each prompt length are increased with 0.5G, 1.3G, 2.7G, respectively. Given the recall and FLOPs in Fig. 4, we set the prompt length to 32 in our experiments.

**Different Ways of Text Prompts Generation.** Since our sentence-level prompts are generated from both reference image and relative caption, we compare two variants, *i.e.,* $\text{Ours}_{RC}$ that the text prompts are generated from the relative caption, and $\text{Ours}_{RI}$ that the text prompts are generated from image, to demonstrate the effectiveness of our method. We record the average recalls of those methods on the two datasets in Table 4. As can be seen, both the $\text{Ours}_{RC}$ and $\text{Ours}_{RI}$ are perform worse than our method across two different datasets. Such results support that our sentence-level prompt is effective in CIR, as it considers both relative caption and reference image.

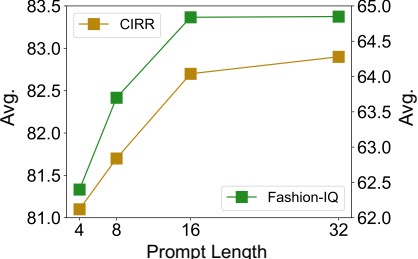

Figure 4: **Ablation studies** in terms of average recalls with regards to different values of *prompt length*.

**Analysis of Alignment Loss.** We record the average recalls using different values of $\gamma$ on the two different datasets in Fig 5. As can be seen, with the increase in $\gamma$ values, the average recall gradually increases across the two datasets, indicating that the additional branch for alignment the text prompt is critical for CIR. Additionally, we can observe from this figure that when $\gamma = 0.8$ our method obtains the highest results, while it drops when the $\gamma$ values $> 0.8$ because overemphasizing text captions to guide sentence-level prompt generation will lose the visual information of the reference image.

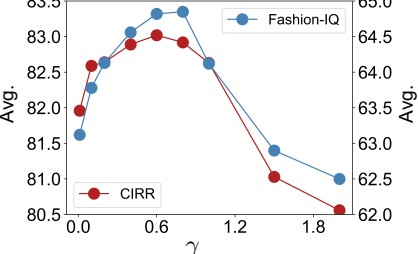

Figure 5: **Analysis** of alignment loss in terms of average recalls with regards to different values of $\gamma$.

## 5 CONCLUSION AND DISCUSSION

In this work, we presented an effective solution for addressing the limitations of prior CIR methods, *i.e.,* late fusion and pseudo-word tokenization approaches. To accomplish this, we harness the capabilities of pre-trained V-L models, such as BLIP-2, to generate sentence-level prompts. By seamlessly integrating these learned sentence-level prompts with relative captions, we enhance CIR performance using well-established text-based image retrieval models. While SPRC has achieved satisfactory results and outperformed previous work on this task, there are still some limitations. When it comes to changes in view angle of objects in reference images, SPRC leads to less than ideal retrieval results, see *Suppl.* in Fig. 7 for details. We empirically find that this problem is closely related to the knowledge acquired by pre-trained models, BLIP-2. Therefore, injecting knowledge about angle transformations into the CIR model to distinguish between different view angles and perspectives holds significant potential.

ACKNOWLEDGMENTS

This work was supported by the Agency for Science, Technology, and Research (A*STAR) through its AME Programmatic Funding Scheme Under Project A20H4b0141, the National Research Foundation (NRF) Singapore under its AI Singapore Programme (AISG Award No: AISG2-TC-2021-003), the Agency for Science, Technology, and Research (A*STAR) through its RIE2020 Health and Biomedical Sciences (HBMS) Industry Alignment Fund Pre-Positioning (IAF-PP) (grant no. H20C6a0032), and partially supported by A*STAR Central Research Fund "A Secure and Privacy-Preserving AI Platform for Digital Health".

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

# A APPENDIX

## CONTENTS

The following items are included in our supplementary material:

## A.1 ADDITIONAL QUALITATIVE ANALYSIS.

We provide additional qualitative results with respect to the Recallsubset@1 metric on two datasets, *i.e.,* (a) **CIRR** and (b) **Fashion-IQ**, in Fig. 6. The results are consistent with our results in main paper, *i.e.,* our method preserves the highest rank of the target images, indicating that our method is robust to complex scenarios where multiple objects are involved in the reference image and intricate modifications like object removal or attribute modification.

## A.2 ANALYSIS OF FAILURE CASES.

Our method has shown outstanding performance as it deals with complex changes in the reference image. When the relative caption involves changes in the view angle of objects in the reference image, the images retrieved by SPRC often have the correct objects but with angles that do not align with what the relative caption describes, which can be seen in Fig. 7. As illustrated in the second case of Fig. 7, the relative caption requires a laptop with a different view angles compared to the reference image. However, the image with the highest recalls displays the same angle as the reference image.

To explore effective solutions to this issue, we constructed various relative captions at different view angles in Fig. 8. After extracting their features using BLIP-2 (Li et al., 2023), we calculated their similarity to the target image. Surprisingly, despite explicitly modifying the angles of the objects in the relative captions, their similarity to the target image remained nearly unchanged. Furthermore, our investigation of the training data of BLIP-2 (Li et al., 2023) revealed that it rarely included variations in these angles. Consequently, the lack of angle-related knowledge in BLIP-2 (Li et al., 2023) is a key factor contributing to the marginal performance gains achieved by our approach.

## A.3 ANALYSIS OF THREE DIFFERENT CIR MECHANISMS ON FASHION-IQ.

Here, we summarized the recalls of three different CIR mechanisms, *i.e.,* late fushion, textual inversion, and text prompt-based method, under different retrieval modals, CLIP (Radford et al., 2021), BLIP (Li et al., 2022), and BLIP-2 (Li et al., 2023), on the **Fashion-IQ** dataset. As shown in Table 5, the late fusion mechanism underperforms Text inversion, and Text inversion falls short compared to Ours+BLIP-2 (Full.), as evident from the consistent results observed across three variants on CLIP, BLIP, and BLIP-2. These findings align with the results presented in Table 3 for the **CIRR** dataset. Furthermore, when employing the same retrieval model, *i.e.,* BLIP-2, the variant methods, *i.e.,* BLIP-2-4cir+Bi (Sum), and Pic2Word$_{su}$+BLIP-2 consistently lower our method across all metrics. This highlights the significance of dynamic learning of sentence-level prompts tailored to relevant captions, as mere acquisition of fixed text prompts appears insufficient for CIR, ultimately leading to enhanced performance.

## A.4 DETAILED COMPARISON OF DIFFERENT VALUES OF $\gamma$.

In Table 6 and Table 7, we record the detailed comparison of our method with different values of $\gamma$ on both **CIRR** and **Fashion-IQ** datasets. As can be seen, our method yields the best **Avg.** recalls when $\gamma = 0.6$ on the **CIRR** dataset and $\gamma = 0.8$ on the **Fashion-IQ** dataset, *i.e.,* **83.02** and **64.85**. Concretely, on the **CIRR** dataset, our method obtains the highest recalls almost across all the metrics

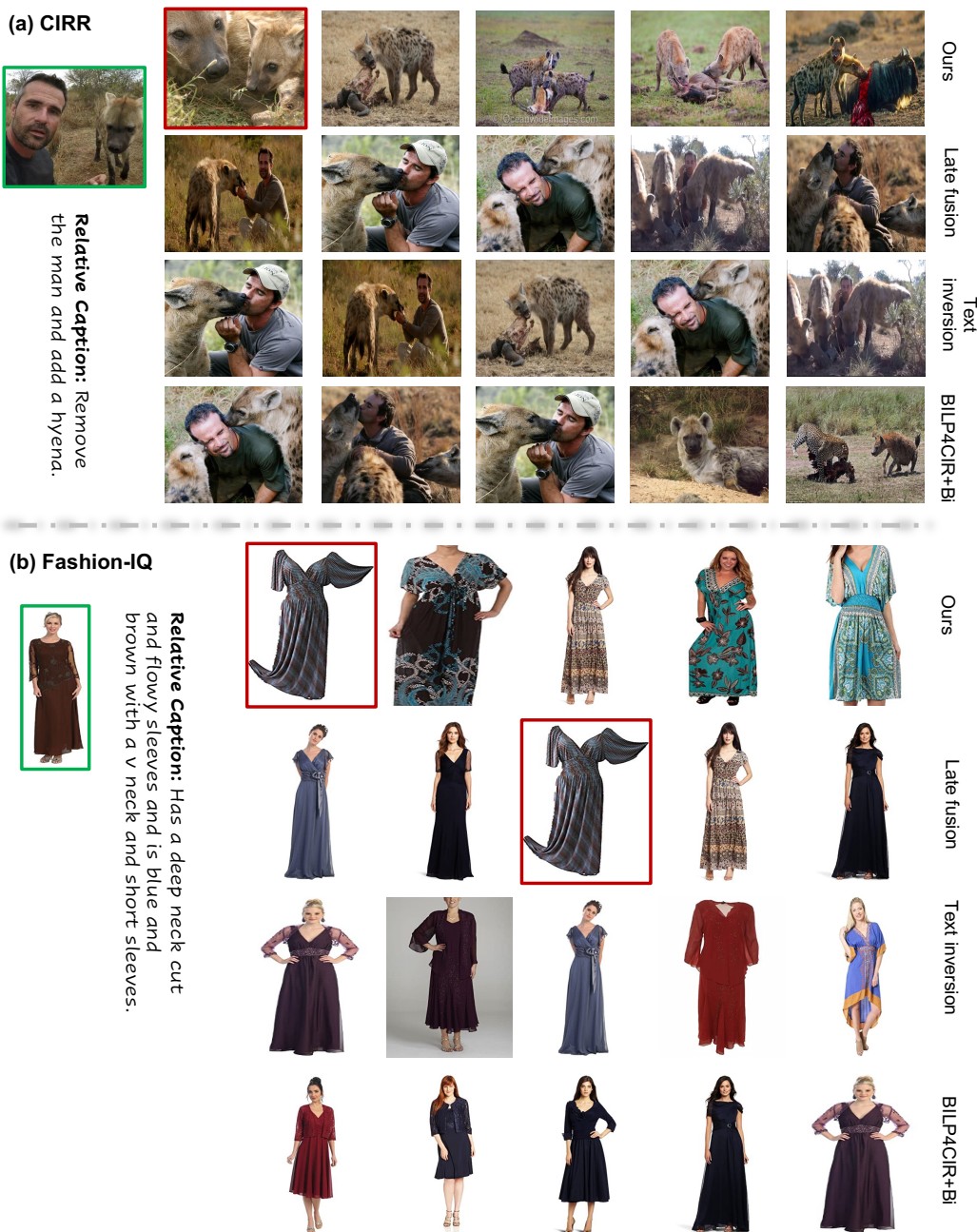

Figure 6: **Additional qualitative** analysis of three different CIR mechanisms with respect to the Recallsubset@1 metric on two datasets, *i.e.,* (a) **CIRR** and (b) **Fashion-IQ**, where the reference image and ground-truth retrieval images are highlighted with green and red outline. Please note that, owing to limitations in available space, we have adjusted the dimensions of specific candidate images to achieve a visually appealing layout.

when $\gamma = 0.6$. On the **Fashion-IQ** dataset, our method obtains 4 best recalls and 4 second-best recalls, thereby yielding the highest average performance, *i.e.,* **64.85**. Notably, our method yields consistent performance regarding different values of $\gamma$, indicating the robustness of our method.

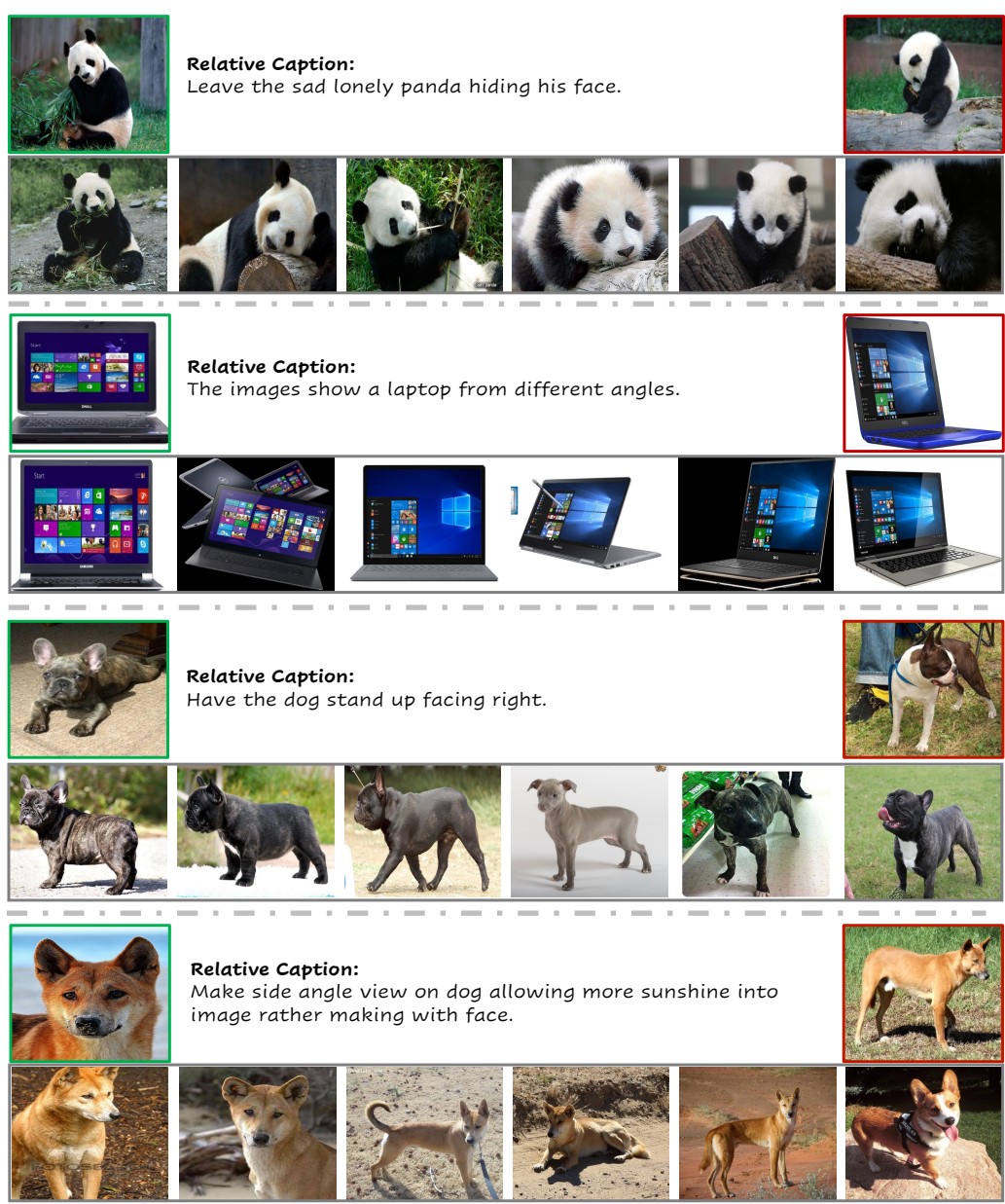

Figure 7: **Failure cases** of our method, where the reference image, target image, and retrieval images are highlighted with green, red, and gray outline. One can see that the relative captions in these cases require modifying the angle of the target object in the reference image.

## A.5 DETAILED COMPARISON OF DIFFERENT LENGTHS OF PROMPT.

Here, we record the detailed recalls with regards to different values of prompt length on both the **CIRR** and **Fashion-IQ** datasets in Table 8 and 8. As can be seen from these tables, our method obtains the highest recalls almost across all the metrics with a prompt length of 32 on the two datasets. Nonetheless, the performance of our method shows obvious variation when the prompt length is set to either 4 and 8 *vs.* 16 and 32, while there is little variation with the prompt length of 16 and 32. Consequently, this indicates that our method begins to reach a plateau once the prompt length reaches 16.

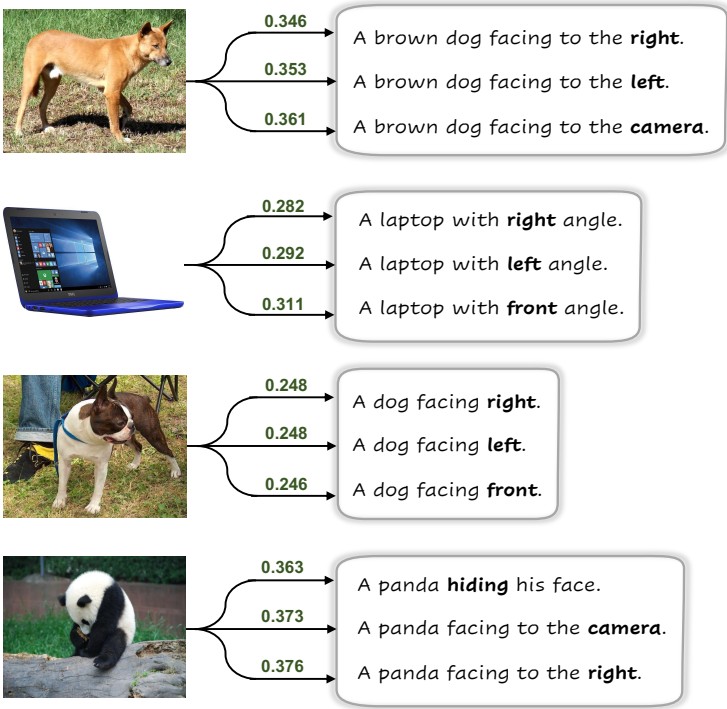

Figure 8: **Similarities** between the target image and relative captions with different view angles. Here, BLIP-2 is adopted to compute the similarity. One can see that BLIP-2 performs limited in distinguishing different view angles.

Table 5: **Ablation** studies with regard to three different CIR mechanisms on **Fashion-IQ** dataset, *i.e.,* (1) Late fusion: CLIP4CIR (Sum) (Baldrati et al., 2022b), BLIP4CIR (Sum), and BLIP-2-4CIR (Sum); (2) Textual inversion: variant Pic2Word (Saito et al., 2023) to supervised method with different retrieval modals, *i.e.,* CLIP, BLIP, and BLIP-2; (3) Text prompt-based method: BLIP4CIR+Bi (Liu et al., 2023a).

| Method | CLIP | BLIP | BLIP-2 | Dress R@10 | Dress R@50 | Shirt R@10 | Shirt R@50 | Toptee R@10 | Toptee R@50 | Average R@10 | Average R@50 | Avg. |
|---|---|---|---|---|---|---|---|---|---|---|---|---|
| CLIP4CIR (Sum) | ✓ | — | — | 31.14 | 55.18 | 35.77 | 57.41 | 38.09 | 61.04 | 35.00 | 57.88 | 46.44 |
| BLIP4CIR (Sum) | — | ✓ | — | 37.13 | 62.67 | 35.92 | 60.40 | 43.60 | 68.28 | 38.88 | 63.78 | 51.33 |
| BLIP-2-4CIR (Sum) | — | — | ✓ | 44.67 | 68.47 | 53.34 | 74.86 | 50.44 | 70.80 | 49.48 | 71.38 | 60.43 |
| Pic2Word$_{su}$+CLIP | ✓ | — | — | 29.25 | 54.53 | 34.10 | 55.39 | 39.67 | 62.67 | 34.34 | 57.53 | 45.94 |
| Pic2Word$_{su}$+BLIP | — | ✓ | — | 36.54 | 60.53 | 40.68 | 62.86 | 44.82 | 66.09 | 40.68 | 63.16 | 51.92 |
| Pic2Word$_{su}$+BLIP-2 | — | — | ✓ | 44.27 | 68.91 | 51.32 | 71.30 | 55.84 | 76.13 | 50.48 | 72.11 | 61.29 |
| BLIP4CIR+Bi | — | ✓ | — | 42.09 | 67.33 | 41.76 | 64.28 | 46.61 | 70.32 | 43.49 | 67.31 | 55.40 |
| Ours+CLIP | ✓ | — | — | 32.72 | 58.01 | 35.92 | 57.75 | 42.89 | 66.19 | 37.18 | 60.65 | 48.91 |
| Ours+BLIP | — | ✓ | — | 42.49 | 65.94 | 42.98 | 65.31 | 48.60 | 71.09 | 44.69 | 67.44 | 56.08 |
| Ours+BLIP-2 (Full.) | — | — | ✓ | **49.18** | **72.43** | **55.64** | **73.89** | **59.35** | **78.58** | **54.92** | **74.97** | **64.85** |

Table 6: **Detailed comparison** of alignment loss in our method in terms of average recalls with regards to different values of $\gamma$ on the **CIRR** dataset. The best and second-best results are marked in **bold** and underlined, respectively.

| | Recall@K | | | | Recall$_s$@K | | | Avg. |
|---|---|---|---|---|---|---|---|---|
| $\gamma$ | **K=1** | **K=5** | **K=10** | **K=50** | **K=1** | **K=2** | **K=3** | |
| 0 | 53.11 | 82.92 | 90.39 | 98.17 | 80.10 | 92.53 | 96.70 | 81.51 |
| 0.01 | 53.18 | 83.29 | 90.09 | 97.84 | 80.52 | 92.95 | 96.66 | 81.91 |
| 0.1 | **54.89** | **84.76** | **91.48** | 97.99 | 80.43 | 92.92 | 97.05 | 82.59 |
| 0.2 | 53.95 | 83.69 | 91.13 | 97.92 | **81.63** | 92.89 | 97.08 | 82.64 |
| 0.4 | 54.43 | 84.34 | 90.69 | 97.79 | 81.43 | 92.99 | **97.34** | 82.89 |
| 0.6 | 54.39 | **84.76** | 91.25 | 97.99 | 81.27 | **93.30** | 97.20 | **83.02** |
| 0.8 | 55.29 | 84.26 | 90.93 | **98.15** | 81.57 | 93.08 | 97.01 | 82.92 |
| 1.0 | 54.43 | 84.07 | 90.76 | 97.96 | 81.20 | 92.77 | 96.91 | 82.63 |
| 1.5 | 53.36 | 82.69 | 90.52 | 97.89 | 79.38 | 91.82 | 96.72 | 81.03 |
| 2.0 | 52.78 | 82.60 | 90.14 | 97.75 | 78.52 | 91.84 | 96.50 | 80.56 |

Table 7: **Detailed comparison** of alignment loss in our method in terms of average recalls with regards to different values of $\gamma$ on the **Fashion-IQ** dataset. The best and second-best results are marked in **bold** and underlined, respectively.

| | Dress | | Shirt | | Toptee | | Average | | |
|---|---|---|---|---|---|---|---|---|---|
| $\gamma$ | **R@10** | **R@50** | **R@10** | **R@50** | **R@10** | **R@50** | **R@10** | **R@50** | **Avg.** |
| 0 | 46.74 | 70.25 | 53.72 | 73.15 | 56.86 | 77.01 | 52.44 | 73.47 | 62.96 |
| 0.01 | 46.55 | 70.30 | 53.53 | 73.06 | 57.32 | 78.22 | 52.47 | 73.86 | 63.16 |
| 0.1 | 48.09 | 71.49 | 54.07 | 73.69 | 57.06 | 78.28 | 53.08 | 74.49 | 63.78 |
| 0.2 | 49.13 | 71.64 | 54.66 | 74.28 | 57.62 | 77.97 | 53.81 | 74.63 | 64.21 |
| 0.4 | **49.92** | 72.24 | 54.96 | 73.55 | 58.23 | 78.48 | 54.37 | 74.76 | 64.56 |
| 0.6 | 48.14 | 72.38 | **56.23** | 74.38 | 58.34 | **79.52** | 54.06 | 75.34 | 64.82 |
| 0.8 | 49.18 | **72.43** | 55.64 | 73.89 | **59.35** | 78.58 | **54.92** | **74.97** | **64.85** |
| 1.0 | 47.74 | 71.24 | 55.05 | **74.43** | 57.88 | 77.82 | 53.56 | 74.50 | 64.03 |
| 1.5 | 46.21 | 70.50 | 52.85 | 72.42 | 57.57 | 77.92 | 52.21 | 73.61 | 62.91 |
| 2.0 | 46.60 | 69.03 | 54.22 | 72.23 | 56.14 | 76.75 | 52.32 | 72.66 | 62.49 |

Table 8: **Detailed comparison** of our method in terms of average recalls with regards to different values of *prompt length* on the **CIRR** dataset. The best and second-best results are marked in **bold** and underlined, respectively.

| | Recall@K | | | | Recall$_s$@K | | | Avg. |
|---|---|---|---|---|---|---|---|---|
| **Length** | **K=1** | **K=5** | **K=10** | **K=50** | **K=1** | **K=2** | **K=3** | |
| 4 | 54.01 | 83.02 | 90.17 | 97.97 | 79.58 | 91.99 | 96.58 | 81.13 |
| 8 | 53.81 | 83.28 | 90.12 | 97.79 | 80.12 | 92.32 | 96.91 | 81.70 |
| 16 | **54.87** | 84.26 | 90.73 | **98.25** | **81.71** | 93.12 | 97.06 | 82.97 |
| 32 | 54.39 | **84.76** | **91.25** | 97.99 | 81.27 | **93.30** | **97.20** | **83.02** |

Table 9: **Detailed comparison** of our method in terms of average recalls with regards to different values of *prompt length* on the **Fashion-IQ** dataset. The best and second-best results are marked in **bold** and underlined, respectively.

| | Dress | | Shirt | | Toptee | | Average | | |
|---|---|---|---|---|---|---|---|---|---|
| **Length** | **R@10** | **R@50** | **R@10** | **R@50** | **R@10** | **R@50** | **R@10** | **R@50** | **Avg.** |
| 4 | 46.01 | 68.69 | 53.63 | 72.47 | 56.60 | 76.89 | 52.08 | 72.78 | 62.43 |
| 8 | 47.65 | 71.29 | 54.02 | 73.45 | 57.37 | 78.48 | 53.01 | 74.41 | 63.71 |
| 16 | 47.99 | **72.68** | **56.28** | 73.23 | 58.59 | **79.04** | 54.29 | **75.32** | 64.81 |
| 32 | **49.18** | 72.43 | 55.64 | **73.89** | **59.35** | 78.58 | **54.92** | 74.97 | **64.85** |

