# OpenReview forum: "Sentence-level Prompts Benefit Composed Image Retrieval"
_ICLR.cc/2024/Conference — ICLR 2024 spotlight_

### Official Review · Reviewer_NuL6 · 2023-10-31

**Soundness:** 3 good
**Presentation:** 3 good
**Contribution:** 2 fair
**Rating:** 6
**Confidence:** 4

**Summary:**

This paper proposes to learn sentence-level prompts for supervised composed image retrieval task, to handle complex changes in CIR task such as modifications involving multiple objects. The sentence-level prompts are generated from query image and relative caption, to yield precise descriptions of specific elements in the query image that are described in the relative caption. Experimental results demonstrate that the proposed method achieves better results on two public CIR benchmarks including FashionIQ and CIRR.

**Strengths:**

1. It is reasonable to generate sentence-level prompts depending on both reference image and relative caption to enrich the expressivity.
2. Moreover, the experiments are solid since the authors conduct a thorough comparison with previous methods.
3. The paper is well-written and the idea is easy to follow.

**Weaknesses:**

1.	in my comprehension, the sentence-level prompts are actually latent vectors output from the MLP layer, so it is hard to make sure the prompts work as expected as demonstrated in Figure 1(c), i.e., decoupling the multiple objects and attributes of query image, and correctly integrating the process of object removal or attribute modification.
2.	It is difficult to understand the pi’ in prompt alignment loss. Whether each reference image has an auxiliary text prompt? As a result, the Figure 2(a) involves two training stages (ITC loss to optimize p and prompt alignment loss to optimize pi’)? Furthermore, during the optimization of pi’, the text encoder is frozen, so the image encoder learns to align with the frozen text encoder; while in optimization of pi, the text encoder is not frozen, so the image encoder learns to align with the updated text encoder. I find it hard to understand how the prompt alignment loss works and it seems very tricky to achieve a good performance.

**Questions:**

The same as weaknesses.

---

> ### Author Response · Authors · 2023-11-17
>
> We thank the reviewer for his/her constructive comments and provide our point-wise replies as follows.
>
> >**W1:** How does sentence-level prompts work like in Fig.1(c).
>
> **A1:** Yes, our sentence-level prompts are latent word embeddings. Nevertheless, the sentence-level prompts in our method are a sequence of word embeddings, although not in human-readable form, which is continuous and beneficial for learning. While the purpose of Fig. 1 (c) is to visualize our motivation.
>
> To further prove whether our method can achieve the effect shown in Fig. 1 (c), we trained SPRC on CIRR and the zero-shot tests on the GeneCIS data demonstrate the efficacy of SPRC in capturing and modifying objects and attributes, i.e., the metrics of Change Attribute, focus attribute, and Change Object in the table below, etc.
>
>
> |                | Focus Attribute | Change Attribute | Focus Object     | Change Object    | Avg. |
> |----------------|-----------------|------------------|------------------|------------------|------|
> |                | R@1,R@2,R@3     | R@1,R@2,R@3      | R@1,R@2,R@3      | R@1,R@2,R@3      |  R@1  |
> | Combiner       | 15.1,27.7,39.8  | 12.1, 22.8, 31.8  | 13.5, 25.4, 36.7 | 15.4, 28.0, 39.6 | 14.0 |
> | SPRC          | 19.9,32.6,42.9  | 16.1, 28.0, 37.7  | 21.8, 35.1, 44.8 | 25.0, 38.5, 48.1 | 20.7 |
> *Table 1: GeneCIS*
>
> >**W2:** How the prompt alignment loss works.
>
> **A2:** Sorry for the confusion. We use the same auxiliary text prompt for all reference images, therefore, '$p_i^\prime$' should be '$p^\prime$'. We have revised it in our revised manuscript.
>
> In our method, two types of loss functions, i.e., ITC loss and alignment loss, are utilized to jointly optimize the sentence-level prompt $p_i$.
>
> Specifically, the alignment loss indicates calculating the L2 distance between $p_i$ and $p'$.
>
> We note that during the alignment of $p_i$ to $p'$, the gradient of $p'$ does not need to be computed.
>
> During the optimization of  $p'$, the text encoder related to $p'$ is not entirely frozen but is aligned through the text encoder trained by p using an EMA approach. At this stage, the image encoder remains frozen. Therefore, during the optimization of $p'$, we only need to learn $p'$, as the text encoder copies weights via EMA, and the image encoder is frozen.
> Hence, the optimization of $p'$ does not affect any parameters of the text encoder, image encoder.
>
> In summary, the primary role of $p'$ is to act as a constraint for the sentence-level prompt $p_i$, ensuring that $p_i$ can align more closely with the word embedding feature space.
>
> To verify the effect of the alignment loss, we compared the performance of the model on the CIRR and F-IQ validation sets in Tables 6 and 7 of the supplementary material. The results show that the smaller the weight of the alignment loss, the worse the performance of the model. This further verifies the effectiveness of the alignment loss, and we also provide the performance without the alignment loss as follows:
>
> |                    | Recall@k k=1 | Recall@k k=5 | Recall@k k=10 | Recall@k k=50 | Recall sub@k k=1 | Recall sub@k k=2 | Recall sub@k k=3 | Avg.  |
> |--------------------|--------------|--------------|---------------|---------------|------------------|------------------|------------------|-------|
> | w/o alignment loss | 53.11        | 82.92        | 90.39         | 98.17         | 80.10            | 92.53            | 96.70            | 81.51 |
> | w/ alignment loss  | 54.39        | 84.76        | 91.25         | 97.99         | 81.27            | 93.30            | 97.20            | 83.02 |
> *Table2: Alignment loss on CIRR dataset*
>
> |                    | Dress R@10 | Dress R@50 | Shirt R@10 | Shirt R@50 | Toptee R@10 | Toptee R@50 | Average R@10 | Average R@50 | Avg.  |
> |--------------------|------------|------------|------------|------------|-------------|-------------|--------------|--------------|-------|
> | w/o alignment loss | 46.74      | 70.25      | 53.72      | 73.15      | 56.86       | 77.01       | 52.44        | 73.47        | 62.96 |
> | w/ alignment loss  | 49.18      | 72.43      | 55.64      | 73.89      | 59.35       | 78.58       | 54.92        | 74.97        | 64.85 |
> *Table2: Alignment loss on F-IQ dataset*

---

> > ### Comment · Reviewer_NuL6 · 2023-11-22
> >
> > Thanks for the authors' response. In the response of A2, I am still confused about the whole learning process. Specifically, how many steps are used to train p', and how do you switch the processes of training backbones and p'? It will be better to give a detailed training algorithm.

---

> > > ### Author Response · Authors · 2023-11-22
> > >
> > > Thanks for your reply. Please see below the response to specific questions.
> > >
> > > >**Q1:** how many steps are used to train $p'$.
> > >
> > > **A1:**
> > > The training of $p'$ only involves ***one step***, utilizing Equation (3) from the main manuscript, which can also be re-formulated as a loss function as follows:
> > > $$
> > > \mathcal{L}_{p'}=-\frac{1}{\mathcal{B}} \sum\_{i \in \mathcal{B}} \log \frac{\exp \left(\tau \boldsymbol{v}_i^T \boldsymbol{u'}\_i\right)}{\sum\_{j \in \mathcal{B}} \exp \left(\tau \boldsymbol{v}\_j^T \boldsymbol{u'}\_i\right)}~.
> > > $$
> > > >**Q2:** how do you switch the processes of training backbones and $p'$.
> > >
> > > **A2:** In each training iteration, the input image and text will go through forward pass twice: once for backbones and once for $p'$. However, we perform only one time backward propagation per training iteration for joint optimization of both backbones and $p'$ by combining their losses.
> > >
> > > We will release our code soon for further clarity.

---

> > > ### Author Response · Authors · 2023-11-23
> > >
> > > Thanks again for your question, we have re-clarified the mentioned concerns in the revised manuscript(in blue color) and we greatly appreciate receiving your further feedback, along with any concerns you might have. We are eager and ready to address them.

---

> > > > ### Comment · Reviewer_NuL6 · 2023-11-23
> > > >
> > > > Thanks for the authors' further response. They have addressed most of my concerns. Therefore, I increase the final rating.

---

> ### Author Response · Authors · 2023-11-23
> **We thank the reviewer again for the valuable feedback and happy to address any remaining concerns**
>
> As the discussion period draws to a close soon, we extend our sincere gratitude to the reviewer for his/her valuable time and insightful feedback. We value the constructive feedback and hope that our responses have appropriately addressed all the concerns.
>
> We really appreciate the valuable time to respond to our feedbacks based on the reviewer's comments. Further, we are happy to address any remaining concerns.

---

### Official Review · Reviewer_a9pf · 2023-10-31

**Soundness:** 3 good
**Presentation:** 3 good
**Contribution:** 3 good
**Rating:** 8
**Confidence:** 5

**Summary:**

The authors propose a sentence prompt generation based approach to composed image retrieval or CIR. They train their system to learn the generation of such sentence prompts for known target images (in the training set) through two innovative loss functions. Then at inference time their system essentially generates a sentence prompt that along with the original user query enables the user to retrieve the target image with greater accuracy since the sentence prompt has a much more fine-grained description of the target image.

**Strengths:**

1. Comprehensive literature survey and good motivation of the problem.
2. Sound approach based on innovative loss functions.
3. Good results that exceed the state of the art.

**Weaknesses:**

1. The overall innovation could be seen as modest. However, I am open to being convinced otherwise.

**Questions:**

1. How does your approach do across domains? Is it able to adapt to domain shifts in other words?

---

> ### Author Response · Authors · 2023-11-17
>
> We thank the reviewer for his/her constructive comments and provide our point-wise replies as follows.
>
> >**W1:**  Innovation moderate; open to persuasion.
>
> **A1:** The key innovations of our research include:
>
> (1) Our work enhances the expression capability and accuracy of CIR by learning sentence-level prompts from reference images and relative captions. This approach is contrasted with the typical late fusion strategies or pseudo-word token generation in existing models, providing an effective solution for complex scenarios such as object removal and attribute modification in CIR.
>
> (2) The effectiveness of our method is not limited to a specific retrieval model; it exhibits consistent superior performance across different models such as CLIP, BLIP, and BLIP-2.
>
> >**W2:** Cross-domain adaptability of SPRC.
>
> **A2:** To verify the domain generalization ability of SPRC, we conducted zero-shot tests on CIRCO, GeneCIS, ImageNet (consists of ImageNet [1] and ImageNet-R [2]). ImageNet consists of samples from several different domains (cartoons, origami, toys, and sculptures), refer to the table below for details.
>
> The results in the tables below show that the zero-shot performance of our method (pre-trained on CIRR) on the CIRCO is significantly better than the previous zero-shot method SEARLE [3].
>
> Our method also surpasses the previous supervised method Combiner (Baldrati et al., 2022a) on GeneCIS dataset. This clearly shows that our model can adapt to domain shifts.
>
> Similarly, using CIRR training and conducting zero-shot tests on ImageNet also achieved noticeable improvements against Combiner.
>
> Although there is a clear domain difference between Fashion-IQ and other data, our method still obtained considerable performance in zero-shot tests trained on the Fashion-IQ dataset, see Table 1, 3. This result further shows that our model has the ability to handle domain shifts.
>
>
>
> | Model                         | mAP@k=5 | mAP@k=10 | mAP@k=25 | mAP@k=50 |
> |-------------------------------|---------|----------|----------|----------|
> | SEARLE           | 11.68   | 12.73    | 14.33    | 15.12    |
> | SPRC(CIRR)                | 22.86   | 23.63    | 25.56    | 26.55    |
> | SPRC(F-IQ)                | 14.21   | 15.18    | 16.86    | 17.74    |
> *Table 1: CIRCO*
>
> |                | Focus Attribute | Change Attribute | Focus Object     | Change Object    | Avg. |
> |----------------|-----------------|------------------|------------------|------------------|------|
> |                | R@1,R@2,R@3     | R@1,R@2,R@3      | R@1,R@2,R@3      | R@1,R@2,R@3      |  R@1  |
> | Combiner       | 15.1,27.7,39.8  | 12.1, 22.8, 31.8  | 13.5, 25.4, 36.7 | 15.4, 28.0, 39.6 | 14.0 |
> | SPRC.          | 19.9,32.6,42.9  | 16.1, 28.0, 37.7  | 21.8, 35.1, 44.8 | 25.0, 38.5, 48.1 | 20.7 |
> *Table 2: GeneCIS*
>
> |                | Cartoon R@10 | Cartoon R@50 | Toy R@10 | Toy R@50 | Origami R@10 | Origami R@50 | Sculpture R@10 | Sculpture R@50 |
> |----------------|--------------|--------------|---------|---------|--------------|--------------|----------------|----------------|
> | Combiner    | 6.1          | 14.8         | 10.5    | 21.3    | 7.0          | 17.7         | 8.5            | 20.4           |
> | Pic2Word    | 8.0          | 21.9         | 13.5    | 25.6    | 8.7          | 21.6         | 10.0           | 23.8           |
> | SPRC(CIRR)           | 9.8          | 22.9         | 10.6    | 24.5    | 15.4         | 28.0         | 9.8            | 22.6           |
> | SPRC(F-IQ)     | 8.0          | 20.1         | 11.4    | 25.5    | 9.3         | 21.5         | 6.1            | 15.2           |
> *Table 3: ImageNet*
>
>
>
> [1] Deng, Jia, et al. "Imagenet: A large-scale hierarchical image database." 2009 IEEE conference on computer vision and pattern recognition. Ieee, 2009.
>
> [2] Hendrycks, Dan, et al. "The many faces of robustness: A critical analysis of out-of-distribution generalization." Proceedings of the IEEE/CVF International Conference on Computer Vision. 2021.
>
> [3] Baldrati, Alberto, et al. "Zero-Shot Composed Image Retrieval with Textual Inversion." arXiv preprint arXiv:2303.15247 (2023).

---

> ### Author Response · Authors · 2023-11-23
> **We thank the reviewer again for the valuable feedback and happy to address any remaining concerns**
>
> As the discussion period draws to a close soon, we extend our sincere gratitude to the reviewer for his/her valuable time and insightful feedback. We value the constructive feedback and hope that our responses have appropriately addressed all the concerns.
>
> We really appreciate the valuable time to respond to our feedbacks based on the reviewer's comments. Further, we are happy to address any remaining concerns.

---

### Official Review · Reviewer_fBxs · 2023-11-02

**Soundness:** 3 good
**Presentation:** 3 good
**Contribution:** 2 fair
**Rating:** 6
**Confidence:** 3

**Summary:**

The paper proposes an novel approach for the Composed Image Retrieval (CIR) problem. The idea is to generate a sentence ("text prompt") to describe both the input image and the relative caption and it for searching the image database using standard text-to-image retrieval methods. The method is evaluated against 10+ baseline methods against two datasets ( CIRR, Fashion - IQ). The results provide significant top-k recall gains over all the baselines.

**Strengths:**

- The proposed method is technically sound and simple.
- The paper is easy to follow, experiments are thorough along with ablations (such as prompt length, weight in the loss function etc).
- Provides SOTA results against 10+ baselines on two public datasets.
- To be open-sourced.

**Weaknesses:**

- Limited novelty: A recent paper (https://arxiv.org/pdf/2310.09291.pdf) with quite similar methodology and motivations except for a nuanced difference: training-free vs learned sentence level prompts. There is a need for contextualizing these methods together, ideally under the same evaluation framework so that we understand the value of learned sentence level prompts proposed by this paper.
- Experimental setup: CIRR dataset experiments uses a random split of the training dataset as the test set for evaluations. The results should be reported in the official (hidden) test set instead. Otherwise reported numbers are not comparable to other papers published in this area.

**Questions:**

Q1: Could the database image-caption pairs be enriched with the proposed sentence generation method and used for refining the search?
Q2: How easy to extend the proposed method for addressing other problem domains or modalities?

---

> ### Author Response · Authors · 2023-11-17
>
> We thank the reviewer for his/her constructive comments and provide our point-wise replies as follows.
>
> >**W1:** Difference with https://arxiv.org/pdf/2310.09291.pdf.
>
> **A1:** Thank you for sharing the CIReVL paper (Karthik et al, (2023). However, we would like to note that CIReVL is a concurrent work with our SPRC, and the arXiv upload date of this paper, i.e., October 13th, is after the submission deadline for ICLR 2024. As suggested, we performed our SPRC on a series of zero-shot tests on the CIRCO test set for comparison with CIReVL (we also discussed the CIReVL in our revised manuscript). The results are as follows:
>
> | Model                         | mAP@k=5 | mAP@k=10 | mAP@k=25 | mAP@k=50 |
> |-------------------------------|---------|----------|----------|----------|
> | CIReVL ViT-L/14         | 18.57   | 19.01    | 20.89    | 21.80    |
> | SPRC ViT-L/14                 | 22.86   | 23.63    | 25.56    | 26.55    |
> *Table 1: CIRCO*
>
> |                | Focus Attribute | Change Attribute | Focus Object     | Change Object    | Avg. |
> |----------------|-----------------|------------------|------------------|------------------|------|
> |                | R@1,R@2,R@3     | R@1,R@2,R@3      | R@1,R@2,R@3      | R@1,R@2,R@3      | R@1  |
> | CIReVL         | 17.9,29.4,40.4  | 14.8, 25.8, 35.8  | 14.6, 24.3, 33.3 | 16.1, 27.8, 37.6 | 15.9 |
> | SPRC          | 19.9,32.6,42.9  | 16.1, 28.0, 37.7  | 21.8, 35.1, 44.8 | 25.0, 38.5, 48.1 | 20.7 |
> *Table 2: GeneCIS*
>
> |                | Cartoon R@10 | Cartoon R@50 | Toy R@10 | Toy R@50 | Origami R@10 | Origami R@50 | Sculpture R@10 | Sculpture R@50 |
> |----------------|--------------|--------------|---------|---------|--------------|--------------|----------------|----------------|
> | Combiner    | 6.1          | 14.8         | 10.5    | 21.3    | 7.0          | 17.7         | 8.5            | 20.4           |
> | Pic2Word    | 8.0          | 21.9         | 13.5    | 25.6    | 8.7          | 21.6         | 10.0           | 23.8           |
> | CIReVL      | 19.2         | 42.8         | 30.2    | 41.3    | 22.2         | 43.1         | 23.4           | 45.0           |
> | SPRC           | 9.8          | 22.9         | 10.6    | 24.5    | 15.4         | 28.0         | 9.8            | 22.6           |
> *Table 3: ImageNet*
>
> The results in Table 1 show that, when pre-trained on the CIRR dataset and tested for zero-shot performance on the CIRCO dataset, our SPRC significantly outperforms CIReVL across all metrics. The domain differences between the two datasets, CIRR and CIRCO, indicate the generalization ability of our proposed model.
>
> Furthermore, from Table 2, zero-shot test results on the GeneCIS test data show that our SPRC model performs better compared to CLReVL.
>
> Additionally, while CLReVL outperforms our SPRC model when tested using ImageNet data, i.e., Table 3, SPRC outperforms the baseline model Combiner (Baldrati et al., 2022a).
>
> >**W2:** Results of CIRR are val. or test set?
>
> **A2:** Thanks for the comment. We would like to clarify that all our experiments follow the settings of previous CIR works. For example, the results recorded in Table 2 from our main manuscript are all from the **test set** of CIRR, while the validation set is only used for ablation studies. This is consistent with previous CIR works (Liu et al., 2023b, Liu et al., 2023a). We also emphasized this point in the revised manuscript.
>
> >**Q1:** Potential to enrich the caption.
>
> **A3:**  Actually, the sentence-level prompts in SPRC are implicit word embeddings rather than explicit words; hence, SPRC cannot explicitly generate real text descriptions to enrich the dataset. However, SPRC is orthogonal to CIReVL, which can enrich data. For example, the target captions generated from CIReVL may be utilized as supervision to train SPRC's prompts to obtain better generalization abilities.
>
> >**Q2** Extensibility of SPRC to Other domains/modalities.
>
> **A4:**  As shown in Table 3, when we perform zero-shot inference on ImageNet data, our model outperforms the previous SOTA zero-shot methods, i.e., Pic2Word (Saito et al., 2023) on Cartoon and Origami domains and obtains comparable performance on other domains. This showcases our model's ability to address different domains. We also observed that CIReVL outperforms our model, indicating that the explicit text prompts generated by the CIReVL model are beneficial for processing data across different domains. This potentially complements the sentence-level prompts in our SPRC, which can enable SPRC to achieve better performance across various domains.

---

> ### Author Response · Authors · 2023-11-23
> **We thank the reviewer again for the valuable feedback and happy to address any remaining concerns**
>
> As the discussion period draws to a close soon, we extend our sincere gratitude to the reviewer for his/her valuable time and insightful feedback. We value the constructive feedback and hope that our responses have appropriately addressed all the concerns.
>
> We really appreciate the valuable time to respond to our feedbacks based on the reviewer's comments. Further, we are happy to address any remaining concerns.

---

### Author Response · Authors · 2023-11-17

Dear reviewers and meta-reviewers,

We appreciate all reviewers for their valuable comments and suggestions. We've revised our manuscript based on reviewers' comments as follows:

(1) We have reviewed CIReVL in Sec.2 for R1.

(2) We have highlighted the results of Tab. 2 are on the test set of CIRR for R1.

(3) We have re-clarified the auxiliary text prompt in Sec. 3 for R3.

(4) We have added the results of SPRC that without alignment loss in Suppl. of Tab. 6 and Tab. 7.

The changes have been highlighted using blue font in the revised paper. We will release our code and models in the camera-ready version, and please see below our response to each reviewer. If you have any questions or suggestions, please put your comments on OpenReview.

---

### Meta-Review · Area_Chair_x3cE · 2023-12-05

**Metareview:**

This paper proposes a composed image retrieval (CIR) method that replaces the input image with a learned sentence prompt that describes the image. The learned prompt is then combined with the relative caption and used as input for a text-to-image retrieval system. All reviewers found the work interesting, and the experimental validation convincing. The authors clarified some misunderstandings and provided extra results. As a result, reviewers now unanimously recommend acceptance. The AC sees no reason to override this recommendation.

**Justification For Why Not Higher Score:**

The writing is clear and the experiments are extensive but the novelty is mid.

**Justification For Why Not Lower Score:**

See above.

---

### Decision · Program_Chairs · 2024-01-16

Accept (spotlight)